# Association between the *bla*_CTX-M-14_-harboring *Escherichia coli* Isolated from Weasels and Domestic Animals Reared on a University Campus

**DOI:** 10.3390/antibiotics10040432

**Published:** 2021-04-13

**Authors:** Montira Yossapol, Miku Yamamoto, Michiyo Sugiyama, Justice Opare Odoi, Tsutomu Omatsu, Tetsuya Mizutani, Kenji Ohya, Tetsuo Asai

**Affiliations:** 1Department of Applied Veterinary Science, United Graduate School of Veterinary Sciences, Gifu University, Gifu 5011193, Japan; montira.y@msu.ac.th (M.Y.); antyobi.1008@gmail.com (M.Y.); michon@gifu-u.ac.jp (M.S.); wentworthprince@yahoo.com (J.O.O.); kenji.ohya@kkd.biglobe.ne.jp (K.O.); 2Bioveterinary Research Unit, Faculty of Veterinary Sciences, Mahasarakham University, Maha Sarakham 44000, Thailand; 3Research and Education Center for Prevention of Global Infectious Diseases of Animals, Tokyo University of Agriculture and Technology, Tokyo 1838538, Japan; tomatsu@cc.tuat.ac.jp (T.O.); tmizutan20002000@gmail.com (T.M.); 4Education and Research Center for Food Animal Health, Gifu University, Gifu 5011193, Japan

**Keywords:** ESBL, plasmid, recombination, wild and domestic animals

## Abstract

Antimicrobial-resistant (AMR) bacteria affect human and animal health worldwide. Here, CTX-M-14-producing *Escherichia coli* isolates were isolated from Siberian weasels (*Mustela sibirica*) that were captured on a veterinary campus. To clarify the source of bacteria in the weasels, we examined the domestic animals reared in seven facilities on the campus. Extended-spectrum β-lactamase (ESBL)-producing *E. coli* were isolated on deoxycholate hydrogen sulfide lactose agar, containing cephalexin (50 μg/mL) or cefotaxime (2 μg/mL), and were characterized with antimicrobial susceptibility testing, pulsed-field gel electrophoresis (PFGE), replicon typing, and β-lactamase typing analyses. Next-generation sequencing of the ESBL-encoding plasmids was also performed. CTX-M-14 producers isolated from both domestic animals and weasels were classified into six clusters with seven PFGE profiles. The PFGE and antimicrobial resistance profiles were characterized by the animal facility. All CTX-M-14 plasmids belonged to the IncI1 type with a similar size (98.9–99.3 kb), except for one plasmid that was 105.5 kb in length. The unweighted pair group method with arithmetic mean (UPGMA) revealed that the CTX-M-14 plasmid in the weasel isolates might have the same origin as the CTX-M-14 plasmid in the domestic animals. Our findings shed further light on the association of antimicrobial resistance between wild and domestic animals.

## 1. Introduction

The distribution of antimicrobial-resistant (AMR) bacteria is a significant concern worldwide. AMR bacteria in humans are linked to antimicrobial resistance in farm and wild animals [1]. Some wild animals are frequently found in or around human societies and domestic animal facilities [2,3,4] and can transfer AMR bacteria to humans and domestic animals within their environment. Additionally, horizontal gene transfer between bacteria via plasmids can potentially lead to the dissemination and proliferation of AMR bacteria [5]. The transfer of plasmids among several bacterial isolates diversifies the distribution of AMR bacteria.

Bacteria that produce CTX-M-type extended-spectrum β-lactamases (ESBLs) are resistant to third and fourth generation cephalosporins, which are commonly used in modern and veterinary medicine. CTX-M-producing bacteria are distributed worldwide and have been reported in wild animals [2,4,5]. In 2015, we isolated CTX-M-14-producing *Escherichia coli* from wild Siberian weasels (*Mustela sibirica*) present at the Gifu University campus (Gifu Prefecture, Japan). Some studies in Japan have reported the occurrence of several *bla*_CTX-__M_ genes in *E. coli* isolated from the feces of domestic animals, such as *bla*_CTX-M-2_, *bla*_CTX-M-14_, and *bla*_CTX-M-25_ in chicken isolates [6], *bla*_CTX-M-14_ and *bla*_CTX-M-27_ in dog isolates [7], and *bla*_CTX-M-__3_, *bla*_CTX-M-__14_, *bla*_CTX-M-__15_, and *bla*_CTX-M-__55_, in pig isolates [8]. Thus, domestic animals could shed CTX-M-producing bacteria in their feces. As some domestic animals are reared on the campus, wild weasels may be exposed to AMR bacteria from these animals.

Several genotyping methods, such as pulsed-field gel electrophoresis (PFGE) analysis [7,8], have been used to determine ESBL bacterial transmission based on the genetic relatedness among bacteria [2,4]. However, transfer of ESBL plasmids among bacteria may lead to diversity in the genotypes of ESBL-producing bacteria. Classification of the incompatibility (Inc) type of plasmids is helpful to the epidemiology of ESBL plasmids among several bacterial genotypes [9], and the analysis of plasmid genomes can reveal the history of plasmid transfer [10,11].

This study aimed to clarify the linkage between CTX-M-14 plasmid-carrying *E. coli* in weasels and other animals present in the facilities located on campus at the Gifu University. Thus, we examined the characteristics and relatedness of the CTX-M-14-producing bacteria and their plasmids.

## 2. Results

### 2.1. The Infection Rate and Antimicrobial Susceptibility Profiles of ESBL-Producing E. coli in Animal Facilities and Wild Siberian Weasels

We isolated ESBL-producing *E. coli* isolates carrying bla_CTX-M-14_ from weasels (two of nine samples: 22.2%) that were susceptible to non-β-lactam antibiotics. We also isolated ESBL-producing *E. coli* isolates carrying bla_CTX-M-14_ from domestic animals (14 of 69 isolates: 20.3%); these isolates were isolated from dog facility 1 (DF1, 8 of 11 samples: 72.7%), dog facility 2 (DF2, four of ten samples: 40%), and cattle facility (CF, two of seven samples: 28.6%, one positive sample from a Holstein cow and another from a beef cow) (Table 1). The CTX-M-14-producing *E. coli* isolates from CF, DF1, and DF2 were also susceptible to the non-β-lactams, except for two isolates from DF1 and DF2 that harbored bla_TEM-1_ as well as bla_CTX-M-14_ and were resistant to tetracycline-nalidixic acid-ciprofloxacin-chloramphenicol and gentamicin-kanamycin-sulfamethoxazole/trimethoprim, respectively.

**Table 1 antibiotics-10-00432-t001:** Prevalence of CTX-M-14-producing *Escherichia coli* in domestic animals in the Gifu University campus in 2016 and their antimicrobial resistance genes.

Animal Facility (Number of Sampling Location ^a^)	Collection Period	Animal Species (Number of Samples)	Number of CTX-M-14- Positive Samples (Prevalence, %)
Dog facility 1 (1)	June	Dog (11)	8 (72.7) ^b^
Dog facility 2 (2)	December	Dog (10)	4 (40) ^b^
Cattle facility (3)	October	Dairy cow (5) Beef cow (2)	2 (28.6)
Dairy cow facility (4)	October	Dairy cow (18)	0 (0)
Pony facility (5)	June	Pony (4)	0 (0)
Goat facility (6)	August	Goat (14)	0 (0)
Laying hen facility (7)	September	Laying hen (5)	0 (0)
Total number of samples		69	14 (20.3)

^a^ The number of each sampling location has been designated in Figure 1. ^b^
*E. coli* harboring *bla*_CTX-M-14_ and *bla*_TEM-1_ were isolated from one of the eight samples and one of the four samples, respectively.

### 2.2. PFGE Analysis

We observed seven PFGE profiles in the analyzed CTX-M-14-producing *E. coli* isolates and classified them into six clusters (clusters A, B, C, D, E, and F), based on 95% pattern similarity from a dendrogram generated from an unweighted pair group method with arithmetic mean (UPGMA) analysis, due to a minor difference of an isolate in cluster C (Figure 2). Six PFGE profiles from CTX-M-14-producing *E. coli* isolates were from the fourteen positive fecal samples of domestic animals and one PFGE profile was from the two positive fecal samples from weasels.

Two PFGE profiles were found in DF1, three in DF2, and one in CF. However, the clonal relationship of CTX-M-14-producing *E. coli* isolates among the various animal facilities or origins was not analyzed in this study.

### 2.3. CTX-M-14 Plasmid Analysis

A total of six plasmids were selected from the six major PFGE profiles of CTX-M-14-producing *E. coli* isolates from the six clusters. After plasmid-conjugation testing, all transconjugants carrying *bla*_CTX-M-14_ with an IncI1 type plasmid and were susceptible to non-β-lactam antibiotics were selected for plasmid DNA extraction. The six plasmids were p130MS from a weasel (cluster A, sample ID: MS7), p105CF from CF (cluster D, sample ID: BC6), p74DF1 and p80DF1 from DF1 (cluster B, sample ID: D11; and cluster F, sample ID: D12, respectively), and p116DF2 and p123DF2 from DF2 (cluster C, sample ID: D23; and cluster E, sample ID: D27, respectively), as shown in Figure 2 and Figure 3.

After assembly and annotation using the Global Plasmidome Analysis Tool (GPAT), the sizes of five of the six plasmids were found to be confined to a narrow range (98.90–99.26 kb) (Figure 3). The sizes of p80DF1, p130MS, p105CF, p116DF2, and p74DF1 were 98,904 bp, 98,904 bp, 99,001 bp, 99,067 bp, and 99,262 bp, respectively, while p123DF2 was 105,498 bp (Table 2).

GPAT analysis showed that all the plasmid genomes analyzed were IncI1 type. The percentages of average nucleotide identity and similarity of aligned nucleotides of all CTX-M-14 plasmids were 99.98–100% and 89.32–98.07%, respectively. As a query, p74DF1, p80DF1, p105CF, p116DF2, p123DF2, and p130MS had the highest nucleotide alignment with p80DF1, p105CF, p130MS, p130MS, p74DF1, and p105CF, respectively (Appendix A
Appendix A).

#### 2.3.1. Genome Comparisons of the CTX-M-14 Plasmids

After annotating all genes on the sequenced plasmids, the Nucleotide Basic Local Alignment Search Tool (BLASTn) Atlas analysis showed a close similarity in genes among all the plasmids, except for p123DF2, which had a 6940-bp insertion sequence (Figure 4A). Read mapping analysis, using the CLC Genomics Workbench, showed close nucleotide sequence similarity with plasmids, with differences in two regions located (i) around *bla*_CTX-M-14_ (7334 bp) and (ii) within the shufflon system plasmid conjugative transfer pilus tip adhesin gene, *PilV* (*PilV* shufflon; 4439 bp; Figure 4).

The differences in nucleotide sequences around *bla*_CTX-M-14_ were as follows: (i) p123DF2: 59 bp deleted from IS*1380* family transposase (IS*1380*); 113 bp deleted from the gene encoding phosphotransferase system, histidine-containing phosphocarrier protein; and 6940 bp of six genes inserted within a gene encoding pyruvate formate-lyase protein; (ii) p116DF2: 113 bp deleted from the histidine-containing phosphocarrier protein gene; and (iii) p74DF1: 61 bp deleted from the pyruvate formate-lyase gene. The differences in nucleotide sequences within the *PilV* shufflon across all plasmids were observed at the 3′-end of the *PilV* shufflon genes *PilVA* (p74DF1, p80DF1, and p105CF) and *PilVA’* (p130MS, p116DF2, p123DF2), and the nucleotide sequence encoded between the *PilV* shufflon and site-specific recombinase genes.

#### 2.3.2. Relatedness between Six CTX-M-14 Plasmids and Their Interrelatedness with Highly Similar Plasmids

Plasmids p74DF1, p80DF1, p105CF, p130MS, and p116DF2 were the most similar to the CTX-M-1/IncI1 plasmid (GenBank accession number KF787110), with 93% query coverage and 99.63% identity, whereas p123DF2 was also similar to an IncI1 plasmid, but its similar plasmid did not contain any resistance genes (GenBank accession number CP001118) and had 84% query coverage and 98.65% identity. UPGMA analysis showed two clusters of eight plasmids, including a cluster of six CTX-M-14 plasmids used in this study and a cluster of two highly similar plasmids (KF787110 and CP001118; Figure 5A).

The UPGMA phylogenetic network of the CTX-M-14 plasmids used in this study showed that a plasmid of DF1 (p80DF1) was closely related to a plasmid from CF (p105CF). Furthermore, the plasmid from CF (p105CF) shared close relatedness with a group of two plasmids from weasel and DF2 (p130MS–p123DF2) and a group of two plasmids from DF1 and DF2 (p74DF1–p116DF2) (Figure 5B).

## 3. Discussion

This study showed clonal spread of CTX-M-14-producing *E. coli* within each facility on the campus of Gifu University and the linkage of CTX-M-14 plasmids among *E. coli* isolates from wild weasels and domestic animals on the campus. Siberian weasels are an invasive mammalian predator species in Japan and are distributed in the Gifu Prefecture and adjacent prefectures [12]. The Gifu University campus is located in a rural area of the Gifu Prefecture, which is surrounded by paddy fields, cultivated fields, and rivers. Additionally, the campus has several companion and livestock animal facilities. In this study, CTX-M-14-producing *E. coli* were isolated in a limited geographical area and carried by free-moving animals (weasels) (Figure 1). A total of three CTX-M-14-producing *E. coli*-positive facilities (DF1, DF2, and CF) were located within 45 m of each other. However, the PFGE profiles of the isolates from weasels and domestic animals from this study were distinct, suggesting that clonal spread of CTX-M-14-producing bacteria was observed in each facility/origin. The possible lack or failure of biosecurity measures between these facilities in relation with human and animal movements could be the factors responsible for AMR bacterial transfer among these facilities. In addition, wastewater from all facilities was collected by an open drainage system and drained into a main pipeline. Therefore, CTX-M-14 ESBL-encoding plasmids may be transmitted between the bacteria in the limited area of domestic animals after the introduction of ESBL-producing bacteria in each facility.

Some reports have demonstrated closely related plasmids in bacteria with different genotypes [11,13,14]. In this study, CTX-M-14-producing isolates circulating in each animal facility might have harbored a closely related plasmid. The CTX-M-14 plasmids shared identical Inc type, the nearest plasmid sequence type profile with a high percentage of average nucleotide identity, and nucleotide alignment; however, an insertion site in one of the plasmids introduced some differences in plasmid size and other features (Figure 3). The differences in the two genes next to *bla*_CTX-M-14_ and the six-gene insertion of p123DF2 near *bla*_CTX-M-14_ (Figure 4A) were associated with the bacterial phosphotransferase system, which is not directly related to plasmid transfer systems [15]. In contrast, differences in the IS*1380* gene are related to antimicrobial resistance gene transfer between plasmids. Additionally, the diversity of shufflon segment composition observed in this study (Figure 4B) could reflect a random recombination between the inverted recombination sites in the shufflon region [16] and/or shufflon rearrangement by the site-specific recombination of genes located next to the shufflon gene [17]. However, no genes encoding protein–shufflon segments were observed in the plasmid genomes, except for shufflon segment D’ in p116DF2 (Figure 4B). Partial IS*1380* and formate C-acetyltransferase genes, along with the disappearance of shufflon segments, might have been caused by the nucleotide-deletion site. Evolution of plasmids is necessary to ensure their maintenance in a new bacterial host following transfer from another bacterial cell [18]. In this study, UPGMA analyses revealed close relatedness of the plasmids (Figure 5), suggesting circulation of the CTX-M-14 plasmid in the studied region. Although the relatedness among plasmids from DF1 and DF2 was observed because of the same UPGMA cluster (Figure 5), the genomes of plasmids from the weasels and DF2 showed close relatedness due to plasmid evolution by insertion/deletion of nucleotide sequences (Figure 4). Thus, all plasmids had close nucleotide sequence relatedness with differences in the plasmid genome. Although in this study we did not investigate the CTX-M-14 plasmid carriage in other Enterobacteriaceae, there is the possibility that the plasmid can spread to other bacteria in the study location.

AMR bacteria are prevalent in wild animals as well as humans and domestic animals [19]. The wild animal hosts may play a role in maintenance and spread of AMR bacteria and resistance genes. The present results show the linkage between the CTX-M-14 plasmid from domestic animals and wild weasels, although it is not clear whether the transfer of plasmids occurred from domestic animals to wild weasels or vice versa. Antimicrobial drugs are widely used in humans and domestic animals, but not for wild animals. However, wild animals can receive AMR bacteria through the environments close to human activities and animal husbandry and, conversely, transmit the AMR bacteria to humans and domestic animals. In addition, AMR bacteria in wild animals may lead to environmental pollution through their urine and feces. At present, despite low prevalence of AMR bacteria in wild animals in Japan [19], understanding the transmission of AMR bacteria and plasmids among humans, domestic animals, and wild animals is essential for human and animal health. In conclusion, control of the distribution of ESBL-producing bacteria and their ESBL-encoding plasmids is important to prevent the dissemination of ESBL genes to other facilities. In this study, the distribution of CTX-M14-producing bacteria in domestic and wild animals at the Gifu University campus was associated with plasmid transfer.

## 4. Materials and Methods

### 4.1. Sampling and Bacterial Isolation

#### 4.1.1. Samples from Siberian Weasels

In 2015, nine weasels were captured using a humane animal trap door (81.22 cm × 26.92 cm × 30.83 cm) placed at the same location (Figure 1) in May (*n* = 4), June (*n* = 1), October (*n* = 1), November (*n* = 2), and December (*n* = 1). The intestinal contents of the weasels were collected. CTX-M-14-producing *E. coli* from these samples were selected on deoxycholate hydrogen sulfide lactose agar (Eiken Chemical, Tokyo, Japan), containing 50 µg/mL cephalexin (Sigma-Aldrich, St. Louis, MO, USA), and incubated overnight at 37 °C. They were identified at the species level using the API 20E Kit (Sysmex bioMérieux, Tokyo, Japan) following the manufacturer’s instructions.

#### 4.1.2. Samples from Domestic Animals from Animal Facilities

A total of sixty-nine fecal samples were collected between April 2016 and December 2016 from seven animal facilities, including DF1 (*n* = 11), DF2 (*n* = 10), (CF, *n* = 7), and facilities for dairy cows (*n* = 18), ponies (*n* = 4), goats (*n* = 14), and egg-laying hens (*n* = 5) (Figure 1). Individual fecal samples from each animal were collected from DF1, CF, and the dairy cow and goat facilities. In the remaining facilities, the samples of different animals in the same vicinity (kennels, cages, or stands) were pooled and the final number of samples was as follows: ten from DF2, four from the pony facility, and five from the egg-laying hen facility.

DF1, DF2, and CF were open-air facilities located in the same area. DF1 included eleven kennels for healthy mixed breed dogs (one kennel per dog), and all kennels shared the same wastewater drainage system. DF2 included ten kennels for forty healthy beagles (one kennel per four dogs). A total of five kennels of DF2 shared the same wastewater drainage system and walking area. CF was located between DF1 and DF2 and housed two healthy young Japanese black cows in separate enclosures and five healthy Holstein cows in tie stalls. DF1, DF2, and CF shared an equipment storage room (inside CF), a food storage room (inside CF), and a fecal storage barn (located between DF1 and CF). All dogs in DF1 and DF2, and cows in CF, were laboratory animals used by veterinary students and staff for research and education; the students took care of these animals. Dogs in DF1 and DF2 were maintained according to the guidelines on animal research and welfare of Gifu University.

AMR bacteria from the fecal samples were isolated on deoxycholate hydrogen sulfide lactose agar containing 50 µg/mL cephalexin and deoxycholate hydrogen sulfide lactose agar containing 2 µg/mL cefotaxime (CTX; Wako Pure Chemical Industries, Osaka, Japan), and incubated overnight at 37 °C. The isolates were identified at the species level using the API 20E Kit following the manufacturer’s instructions.

### 4.2. Ethics Statement

This study was approved by the Ethics Committee for Animal Research and Welfare of Gifu University (approval number 17,109) and by the Ethics Committee for Academic Research of Captured Animals in Gifu Prefecture (approval number 821).

### 4.3. Antimicrobial Susceptibility Testing

Commercial broth microdilution tests (Eiken Chemical, Tokyo, Japan) were performed to determine the minimum inhibitory concentrations of the following twelve antimicrobial agents: ampicillin (1–128 µg/mL), cefazolin (1–128 µg/mL), CTX (0.5–64 µg/mL), meropenem (0.25–32 µg/mL), gentamicin (0.5–64 µg/mL), kanamycin (1–128 µg/mL), tetracycline (0.5–64 µg/mL), nalidixic acid (1–128 µg/mL), ciprofloxacin (0.03–4 µg/mL), colistin (0.12–16 µg/mL), chloramphenicol (1–128 µg/mL), and sulfamethoxazole/trimethoprim (2.38–152 µg/mL/0.12–8 µg/mL). The resistance breakpoints of all antimicrobial agents, except for colistin, were defined according to the guidelines of the Clinical & Laboratory Standards Institute [20], and the resistance breakpoint of colistin was defined according to the guidelines of the European Committee on Antimicrobial Susceptibility Testing [21] (http://www.eucast.org, accessed on April 13, 2021). ESBL production was confirmed by performing a double-disk synergy test using clavulanate (10 µg/disk)/amoxicillin (20 µg/disk), CTX (30 µg/disk), ceftazidime (30 µg/disk), and cefpodoxime sodium (10 µg/disk) (Nissui Pharmaceutical, Tokyo, Japan) as previously described [22].

### 4.4. β-Lactamase Gene Identification

β-lactamase genes were identified by performing multiplex polymerase chain reaction (PCR) [23], and the subtypes of the CTX-M-9-group β-lactamases were determined using sequencing analysis with previously reported primer pairs [24]. Both strands of the amplified DNA fragments were sequenced at the Life Science Research Center of Gifu University, and the encoded amino acid sequences were analyzed using the BLAST program (National Center for Biotechnology Information [NCBI], Bethesda, MD, USA).

### 4.5. PFGE Analysis

The genotypes of ESBL-producing *E. coli* isolates were analyzed according to the standardized PulseNet PFGE protocol for non-O157 *E. coli* isolates using the CHEF-DR III System (Bio-Rad Laboratories, Hercules, CA, USA); the bacterial genomic DNA was digested with the restriction enzyme *Xba*I (Takara Bio, Shiga, Japan). *Xba*I-digested DNA of the *Salmonella* serotype Braenderup H9812 isolate was used as the universal size standard. The PFGE profiles were analyzed using a UPGMA dendrogram with 1% optimization and 1% band-filtering tolerance with BIONUMERICS software version 7.6.1 (Applied Maths NV, Sint-Martens-Latem, Belgium).

### 4.6. Selected Isolates for Plasmid Sequencing

After PFGE analysis, plasmid characterization was performed. At least one isolate per cluster (based on PFGE analysis) was selected and subjected to plasmid-conjugation using the broth mating method with rifampicin- and nalidixic acid-resistant *E. coli* DH5α as recipients as described previously [25]. The transconjugants were subjected to PCR analysis using primers for the CTX-M-9 group [24], and antimicrobial susceptibility analysis and PCR-based plasmid replicon typing using previously reported primer sets [9] were performed.

A transconjugant isolate with an Inc type plasmid was selected from each cluster for plasmid DNA purification. Briefly, plugs of the transconjugants were incubated with the S1 Nuclease enzyme (Promega, Madison, WI, USA) and subjected to PFGE in a SeaKem Agarose gel (Lonza, Rockland, ME, USA) [26]. After staining the S1-PFGE gel with SYBR Gold Nucleic Acid Stain (Invitrogen, Carlsbad, CA, USA), the plasmid bands were cut with the aid of a blue-light transilluminator and purified using the MonoFas DNA Purification Kit (GL Sciences, Tokyo, Japan). Purified plasmid DNA was sequenced on the MiSeq System (Illumina, San Diego, CA, USA) according to the manufacturer’s instructions using the Nextera XT Library Prep Kit (Illumina), optimized for plasmids. *De*
*novo* assembly was performed using the A5-miseq pipeline [27], followed by manual annotation with the GPAT, available in the database of pathogen genomic and epidemiology (GenEpid-J) (Pathogen Genomic Center, National Institute of Infectious Diseases, Toyama, Japan).

### 4.7. Analysis of Plasmid Sequences

#### 4.7.1. CTX-M-14 Plasmid Similarity Analysis

The plasmid Inc and sequence types were investigated using GPAT. To identify plasmid similarity, BLAST was used to calculate the average nucleotide identity and perform nucleotide alignments of the plasmid sequences [28]. The percentage of aligned nucleotides was calculated as [(number of similar nucleotides between query and reference sequences after alignment)/(number of reference sequences)] × 100. The plasmid genomes were compared by visualization with CLC Genomics Workbench software version 10.0.1 (Qiagen, Hilden, Germany). A BLAST Atlas view was generated using the GView server [29]. Genome mapping analysis was performed using Multiple Alignment of Conserved Genomic Sequence with Rearrangements (Mauve) [30] and the GView program version 1.7 [29].

#### 4.7.2. Relatedness among All Plasmids Used in this Study and Interrelatedness between the Highly Similar Plasmids

All plasmid genomes used in this study were queried for highly similar sequences using Standard Nucleotide BLAST of the NCBI database. The plasmid genomes used in this study, and their most highly similar sequences, were aligned and imported into SplitsTree software version 4.11.3 (SplitsTree4) [31] to construct the UPGMA phylogenetic network for analyzing plasmid relatedness. Thereafter, only the plasmid genome sequences of interest were aligned and imported into SplitsTree4 to construct the UPGMA phylogenetic network for analyzing the relatedness among them.

#### 4.7.3. Nucleotide Sequence Accession Numbers

The complete nucleotide sequences of the p74DF1, p80DF1, p105CF, p116DF2, p123DF2, and p130MS plasmids have been submitted to the GenBank database under accession numbers MK764026, MK764028, MK764025, MK764029, MK764024, and MK764027, respectively.

## Figures and Tables

**Figure 1 antibiotics-10-00432-f001:**
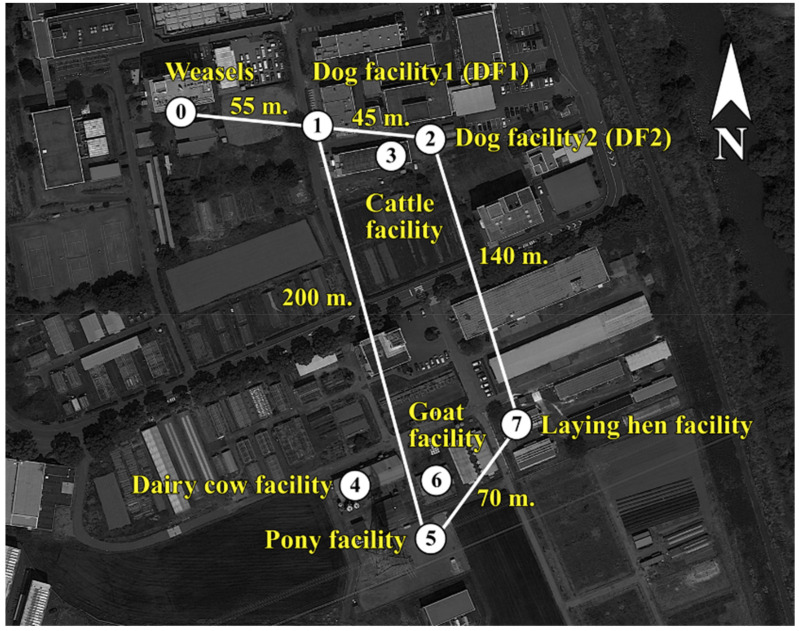
Sampling locations at the Gifu University: ⓪, a trapping location of nine Siberian weasels (*Mustela sibirica*); ①, dog facility 1 (DF1); ②, dog facility 2 (DF2); ③, a cattle facility; ④, a dairy cow facility; ⑤, a pony facility; ⑥, a goat facility; ⑦, an egg-laying hen facility. The distances between the facilities were as follows: locations ⓪ and ①, 55 m; locations ① and ②, 45 m; locations ② and ⑦, 140 m; locations ⑤ and ⑦, 70 m; locations ① and ⑤, 200 m.

**Figure 2 antibiotics-10-00432-f002:**
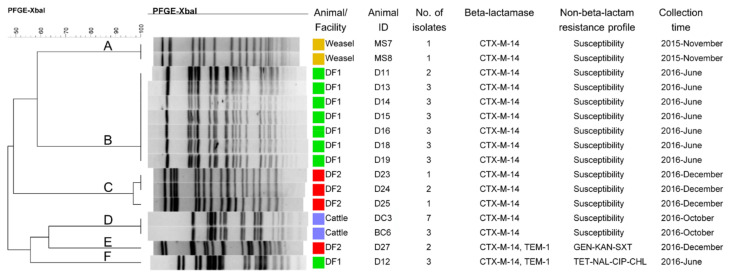
The CTX-M-14-producing *E. coli* isolates from 14 CTX-M-14-producing *E. coli*-positive samples collected from three domestic animal facilities and wild weasels were classified into six clusters (cluster A, B, C, D, E, and F) and seven pulsed-field gel electrophoresis (PFGE) profiles (an unweighted pair group method with arithmetic mean (UPGMA) analysis: 1% optimization and 1% band-filter tolerance). BC, beef cow; CHL, chloramphenicol; CIP, ciprofloxacin; D, dog; DC, dairy cow; DF1, dog facility 1; DF2, dog facility 2; GEN, gentamycin; KAN, kanamycin; NAL, nalidixic acid; MS, *Mustela sibirica* (weasel); SXT, sulfamethoxazole/trimethoprim; TET, tetracycline.

**Figure 3 antibiotics-10-00432-f003:**
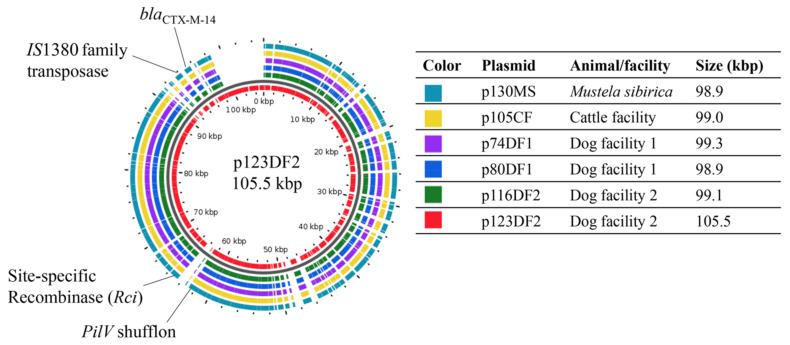
Physical mapping composition of the sequences of six CTX-M-14 plasmids by BLAST Atlas analysis using p130MS, p105CF, p74DF1, p80DF1, and p116DF2 as query sequences against the longest CTX-M-14 plasmid (p123DF2) as reference sequence. Nucleotide sequences were highly similar among the six plasmids, though p123DF1 had an insertion site. This figure was constructed using the GView Server after performing BLAST Atlas analysis.

**Figure 4 antibiotics-10-00432-f004:**
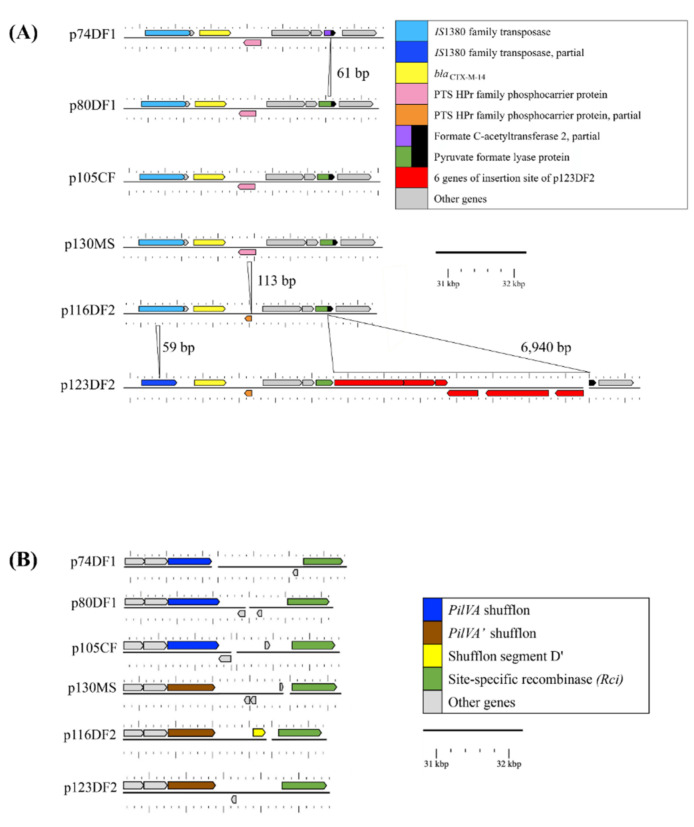
The difference in two regions located around (**A**) *bla*_CTX-M-14_ (7334 bp) and (**B**) within the shufflon system plasmid conjugative transfer pilus tip adhesin *PilV* gene (*PilV* shufflon; 4439 bp). (**A**) Genes around *bla*_CTX-M-14_ differed in terms of the whole/partial presence or absence of (i) a gene encoding IS*1380* family transposase (IS*1380*), (ii) two genes encoding the bacterial phosphotransferase system comprising histidine-containing phosphocarrier protein (HPr) and pyruvate formate-lyase (*PFL*), and (iii) six genes encoding formate C-acetyltransferase, pyruvate formate-lyase 2-activating enzyme, fructose-like phosphotransferase enzyme IIB component 3, putative DNA-binding transcriptional regulator, phosphoethanolamine transferase Cpt A, and phosphoenolpyruvate carboxylase. All plasmids had an insertion sequence element, such as the ISEcp9 family transposase upstream of *bla*_CTX-M-14_. They contained the whole insertion sequence element (identical to ISEcp1 family transposase (1656 bp); deposited under GenBank accession number AJ242809), except for p123DF2, which had a partial IS*1380* sequence due to a 59-bp deletion within its IS*1380* gene (nucleotide positions 1082 to 1139 of AJ242809). Next to *bla*_CTX-M-14_ in four of the six plasmids were 483 nucleotides of the *HPr* gene, while the remaining two plasmids had 183 nucleotides (p116DF2) or 201 nucleotides (p123DF2) of the *HPr* gene. The 113-bp deletion within the *HPr* gene in these two plasmids disrupted the open reading frame (ORF) of the *HPr* gene in each case. At the location of the *PFL* gene, only p123DF2 was disrupted with an insertion sequence (6940 bp) comprising the six genes. Additionally, p74DF1 had a 61 bp deletion within its *PFL* gene. (b) Annotation revealed that a gene encoding the *PilV* shufflon was present in all plasmids and a gene encoding shufflon segment D’ was present next to the *PilV* shufflon gene in p116DF2. Read mapping analysis showed that all plasmids had an identical 1090 bp sequence encoding the N-terminal portion of the *PilV* shufflon. In contrast, the 3′-end encoded by the *PilV* shufflon gene was represented by shufflon segment A (p105CF, p74DF1, and p80DF1) or A’ (p130MS, p116DF2, and p123DF2). The inverse shufflon segments (A’ or A) were located next to the *PilVA* or *PilVA*’ genes. Only p74DF1 had a 271-bp insertion within the *PilV* shufflon gene, and its ORF was disrupted. Although shufflon segment was not present in all plasmids, Nucleotide Basic Local Alignment Search Tool (BLASTn) analysis and multiple-genome alignment using the Mauve program showed the following divergent patterns of shufflon segment compositions in the plasmids: p105CF, segment A–segment B–segment D–segment C, p130MS, A–B–D–C p74DF1, A–C–B–D; p80DF1, A–C–D–B; p116DF2, A–B–D; and p123DF2, A–D–C–B. p105CF was from the cattle facility; p130MS was from a Siberian weasel (*Mustela sibirica*); p74DF1 and p80DF1 were from dog facility 1 (DF1); and p116DF2 and p123DF2 were from dog facility 2 (DF2).

**Figure 5 antibiotics-10-00432-f005:**
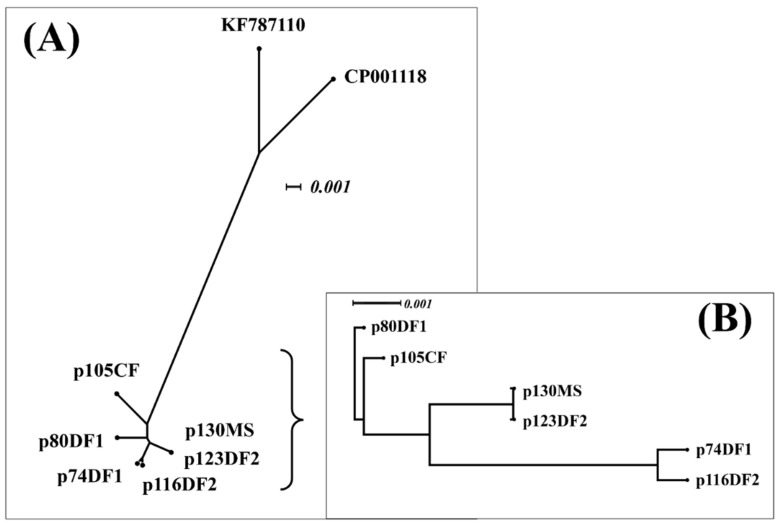
Relatedness of the genomes of the six plasmids in this study. (**A**) An unweighted pair group method with arithmetic mean (UPGMA) phylogram of the six CTX-M-14 plasmids and highly similar plasmid sequences was created using the SplitsTree4 program using uncorrected P characters, UPGMA distances, and equal angle spilt transformations. (**B**) UPGMA phylogram of the six CTX-M-14 plasmids and a highly similar plasmid was created using the SplitsTree4 program using uncorrected P characters, UPGMA distances, and rooted equal angle transformation.

**Table 2 antibiotics-10-00432-t002:** Characteristics of six donor isolates carrying CTX-M-14 plasmids and their transconjugants based on pulsed-field gel electrophoresis (PFGE) profiles.

Animal Origin or Facility	Donor				Transconjugant (TC)			
	Sample ID ^a^	β-Lactamase	Non-β-Lactam-Resistance Profile ^b^	Plasmid ID	β-Lactamase	Non-β-Lactam-Resistance Profile	Replicon Type	Plasmid Size (bp) ^c^
Siberian weasel	MS7	CTX-M-14	Susceptible	p130MS	CTX-M-14	Susceptible	IncI1	98,904
Cattle facility (CF)	BC6	CTX-M-14	Susceptible	p105CF	CTX-M-14	Susceptible	IncI1	99,001
Dog facility 1 (DF1)	D11	CTX-M-14, TEM-1	TET, NAL, CIP, CHL	p74DF1	CTX-M-14	Susceptible	IncI1	99,262
	D12	CTX-M-14	Susceptible	p80DF1	CTX-M-14	Susceptible	IncI1	98,904
Dog facility 2 (DF2)	D23	CTX-M-14	Susceptible	p116DF2	CTX-M-14	Susceptible	IncI1	99,067
	D27	CTX-M-14, TEM-1	GEN, KAN, SXT	p123DF2	CTX-M-14	Susceptible	IncI1	105,498

^a^ BC, beef cow; D, dog; MS, *Mustela sibirica.*
^b^ CHL, chloramphenicol; CIP, ciprofloxacin; GEN, gentamycin; KAN, kanamycin; NAL, nalidixic acid; SXT, sulfamethoxazole/trimethoprim; TET, tetracycline. ^c^ Determined by next-generation sequencing.

## Data Availability

The data in this study can be accessed at www.ncbi.nlm.nih.gov (nucleotide databases) (accessed on 13 April 2021). The accession numbers can be found at: https://www.ncbi.nlm.nih.gov/nuccore/MK764024 (accessed on 13 April 2021), MK764024; https://www.ncbi.nlm.nih.gov/nuccore/MK764025 (accessed on 13 April 2021), MK764025; https://www.ncbi.nlm.nih.gov/nuccore/MK764026.1 (accessed on 13 April 2021), MK764026.1; https://www.ncbi.nlm.nih.gov/nuccore/MK764027 (accessed on 13 April 2021), MK764027; https://www.ncbi.nlm.nih.gov/nuccore/MK764028 (accessed on 13 April 2021), MK764028; and https://www.ncbi.nlm.nih.gov/nuccore/MK764029 (accessed on 13 April 2021), MK764029.

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
