# Peer review of "Association between the *bla*_CTX-M-14_-harboring *Escherichia coli* Isolated from Weasels and Domestic Animals Reared on a University Campus"

_antibiotics, 2021, doi:10.3390/antibiotics10040432_

Round 1
Reviewer 1 Report
General comments:
In this research paper, the authors clarified the linkage between CTX-M-14 plasmid-carrying E. coli in weasels and other animals present in the facilities located on campus at the Gifu University, and further certified that the distribution of CTX-M14-producing bacteria in domestic and wild animals at the Gifu University campus was associated with plasmid transfer. The tests and analyses were carried out well, and the manuscript is generally well-written.
Furtehr comments:
- I suggest change the strain to isolate in the manuscript.
- The CTX-M-14 positive E. coli isolates are universal in the human and veterinary clinic, so the innovativeness of this study is average.
Author Response
Answer to Reviewer 1
General comments:
In this research paper, the authors clarified the linkage between CTX-M-14 plasmid-carrying E. coli in weasels and other animals present in the facilities located on campus at the Gifu University, and further certified that the distribution of CTX-M14-producing bacteria in domestic and wild animals at the Gifu University campus was associated with plasmid transfer. The tests and analyses were carried out well, and the manuscript is generally well-written.
Response
We thank you very much for your general comment.
Further comments:
Comment 1
“I suggest change the strain to isolate in the manuscript.”
Response
Thank you for your comment. We changed “strain to isolate” in line 19 (of abstract part), line 45 (of introduction part), line 74, 76, 77, 96, 141 (of results part), line 255 (of discussion part) and line 361, 362, 365, 369 (of materials and methods).
Comment 2
“The CTX-M-14 positive E. coli isolates are universal in the human and veterinary clinic, so the innovativeness of this study is average.”
Response
We thank you very much for your comment.
Reviewer 2 Report
In this paper, Yossapol et al. use a range of techniques to determine the relatedness of several antimicrobial resistance plasmids occurring in wild weasels and domestic animals on a university campus.
The manuscript is very clear and well-written, with a suitable introduction that covers both the wider problem and the specific study. The authors’ results are presented clearly and with all appropriate details to enable comprehension of the work. Tables and figures seem complete, with informative legends.
The majority of methods given are highly appropriate for the study and are clearly-written. I am not familiar enough with UPGMA or GPAT to comment on those techniques, though I do believe that the analysis in this manuscript has been conducted rigorously. An appropriate ethics statement is also given. The authors have provided their plasmid sequences online, enabling further analysis by other researchers.
The Discussion is appropriate to the study’s findings and raises no points that are particularly controversial. Overall I think the authors’ analysis here is robust and contributes to the field in an interesting way. Whilst the work herein is not of dramatic impact outside of its immediate field, the science is nonetheless robust and complete.
Specific comments:
- Line 75: “non-β-lactams antibiotics” should read “non-β-lactam antibiotics”
- Lines 206-7: The authors state “p123DF2 was also an IncI1 plasmid, but it did not contain any resistance genes”. Does p123DF2 not encode CTX-M-14? Perhaps the authors mean to write “p123DF2 was also homologous to a known IncI1 plasmid, though that plasmid does contain any resistance genes”. Alternatively, I may be misunderstanding the point being made here by the authors.
- Whilst not within the immediate scope of the work, I wonder whether these plasmids are also spreading in the non- coli bacteria in the area. Can the authors comment on whether any other species of resistant enterobacteria were isolated during their study? Perhaps a note on the likelihood of plasmid carriage by other bacterial species in the immediate area could be added to the Discussion.
Author Response
Answer to Reviewer 2
General comment:
In this paper, Yossapol et al. use a range of techniques to determine the relatedness of several antimicrobial resistance plasmids occurring in wild weasels and domestic animals on a university campus.
The manuscript is very clear and well-written, with a suitable introduction that covers both the wider problem and the specific study. The authors’ results are presented clearly and with all appropriate details to enable comprehension of the work. Tables and figures seem complete, with informative legends.
The majority of methods given are highly appropriate for the study and are clearly written. I am not familiar enough with UPGMA or GPAT to comment on those techniques, though I do believe that the analysis in this manuscript has been conducted rigorously. An appropriate ethics statement is also given. The authors have provided their plasmid sequences online, enabling further analysis by other researchers.
The Discussion is appropriate to the study’s findings and raises no points that are particularly controversial. Overall, I think the authors’ analysis here is robust and contributes to the field in an interesting way. Whilst the work herein is not of dramatic impact outside of its immediate field, the science is nonetheless robust and complete.
Response
We thank you very much for your general comment.
Specific comments:
Comment 1
Line 75: “non-β-lactams antibiotics” should read “non-β-lactam antibiotics”
Response
Thanks for your comment. We have revised “non-β-lactams antibiotics” to be “non-β-lactam antibiotics” in line 75.
Comment 2
Lines 206-7: The authors state “p123DF2 was also an IncI1 plasmid, but it did not contain any resistance genes”. Does p123DF2 not encode CTX-M-14? Perhaps the authors mean to write “p123DF2 was also homologous to a known IncI1 plasmid, though that plasmid does contain any resistance genes”. Alternatively, I may be misunderstanding the point being made here by the authors.
Response
Thank you for the useful comments. We would like to explain as your sentence “p123DF2 was also homologous to a known IncI1 plasmid, though that plasmid does contain any resistance genes.” Thus, we revised the sentence in line 206 – 207 to be “whereas p123DF2 was also similar to an IncI1 plasmid, but its similar plasmid did not contain any resistance genes”
Comment 3
Whilst not within the immediate scope of the work, I wonder whether these plasmids are also spreading in the non- coli bacteria in the area. Can the authors comment on whether any other species of resistant enterobacteria were isolated during their study? Perhaps a note on the likelihood of plasmid carriage by other bacterial species in the immediate area could be added to the Discussion.
Response
Thank you for the specific comments. We wonder as your comment.
According to the materials and methods, the 3 – 5 pink, flat, and dry colonies on DHL agar containing 2 µg/mL cefotaxime (CTX-DHL) were picked up to identified bacterial species by API20E and the results showed that all of them were E. coli. The other morphologies of bacterial colonies were on CTX-DHL agar, but we did not pick up them to identified bacterial species because in this study, we aimed to search for the linkage between CTX-M-14-producing E. coli isolates from weasels and other animals in many facilities on university campus by PFGE analysis. Thus, we pointed on E. coli. However, the UPGMA dendrogram did not showed the clonal relationships of CTX-M-14-producing E. coli isolates among domestic animals in many facilities and weasels.
With regard to the likelihood of plasmid carriage in other bacterial species we included a statement “Although in this study we did not investigate the CTX-M-14 plasmid carriage in other Enterobacteriaceae, there is the possibility that the plasmid cab spread to other bacteria in the study location.” in the Discussion (line 278 – 280).